# Correlation between Alzheimer’s Disease and Gastrointestinal Tract Disorders

**DOI:** 10.3390/nu16142366

**Published:** 2024-07-21

**Authors:** Julia Kuźniar, Patrycja Kozubek, Magdalena Czaja, Jerzy Leszek

**Affiliations:** 1Student Scientific Group of Psychiatry, Faculty of Medicine, Wroclaw Medical University, 50-369 Wroclaw, Poland; patrycja.kozubek@student.umw.edu.pl (P.K.); magdalena.czaja@student.umw.edu.pl (M.C.); 2Department of Psychiatry, Faculty of Medicine, Wroclaw Medical University, 50-369 Wroclaw, Poland; jerzy.leszek@umw.edu.pl

**Keywords:** Alzheimer’s disease, microbiome, *Helicobacter pylori*, periodontitis, inflammatory bowel disease, brain–gut–microbiota axis

## Abstract

Alzheimer’s disease is the most common cause of dementia globally. The pathogenesis is multifactorial and includes deposition of amyloid-β in the central nervous system, presence of intraneuronal neurofibrillary tangles and a decreased amount of synapses. It remains uncertain what causes the progression of the disease. Nowadays, it is suggested that the brain is connected to the gastrointestinal tract, especially the enteric nervous system and gut microbiome. Studies have found a positive association between AD and gastrointestinal diseases such as periodontitis, *Helicobacter pylori* infection, inflammatory bowel disease and microbiome disorders. *H. pylori* and its metabolites can enter the CNS via the oropharyngeal olfactory pathway and may predispose to the onset and progression of AD. Periodontitis may cause systemic inflammation of low severity with high levels of pro-inflammatory cytokines and neutrophils. Moreover, lipopolysaccharide from oral bacteria accompanies beta-amyloid in plaques that form in the brain. Increased intestinal permeability in IBS leads to neuronal inflammation from transference. Chronic inflammation may lead to beta-amyloid plaque formation in the intestinal tract that spreads to the brain via the vagus nerve. The microbiome plays an important role in many bodily functions, such as nutrient absorption and vitamin production, but it is also an important factor in the development of many diseases, including Alzheimer’s disease. Both the quantity and diversity of the microbiome change significantly in patients with AD and even in people in the preclinical stage of the disease, when symptoms are not yet present. The microbiome influences the functioning of the central nervous system through, among other things, the microbiota–gut–brain axis. Given the involvement of the microbiome in the pathogenesis of AD, antibiotic therapy, probiotics and prebiotics, and faecal transplantation are being considered as possible therapeutic options.

## 1. Introduction

Neuroinflammation is a common feature of the pathogenic mechanisms involved in various neurodegenerative diseases (NDs) that are incurable, debilitating disorders resulting in progressive degeneration and death of neurons. NDs are a leading cause of global death and disability, and their incidences are expected to increase in the following decades [1]. The central nervous system has been proven to be bi-directionally connected to the gastrointestinal tract, the enteric nervous system (ENS) and the mycobiome via the sympathetic and parasympathetic nerves, the hypothalamic–pituitary–adrenal axis, the immune system, hormones or metabolic signalling. The ENS system, together with the sympathetic and parasympathetic nervous system, is the so-called brain–gut–microbiota axis [2]. NDs are not only confined to the brain as previously believed but might be a product of bidirectional communication of the body with the brain via inter-organ communication [1]. Alzheimer’s disease (AD) is the most common type of dementia, accounting for 60% to 80% of all cases. It is estimated that this disorder affects more than 50 million patients worldwide. The deposition of amyloid-β (Aβ) in the brain parenchyma and the cerebral vasculature, the presence of intraneuronal neurofibrillary tangles and the loss of synapses are neuropathological hallmarks of AD, although it remains unclear what primarily triggers and drives the progression of the disease [3]. Some of the earliest symptoms manifest years before receiving a clinical diagnosis of dementia, including changes in mood (depressive symptoms, apathy), anxiety and sleep. Later-stage symptoms include impaired judgment, disorientation, confusion aggression, agitation, delusions and hallucinations [4]. Several studies have found a positive association between AD and disorders of the intestinal microflora, taking medications for gastritis (*Helicobacter pylori* infection), peptic ulcer disease, inflammatory bowel disease (IBD), and gastroesophageal reflux disease. A likely role for the immune system, intestinal dysbiosis, intestinal amyloid beta (Aβ) accumulation, vagus nerve stimulation, inflammatory processes and lipid metabolism is suggested. Both the quantity and diversity of the microbiome change with age, and significant changes are also seen in AD patients and even people in the preclinical stage of the disease. The number of pro-inflammatory bacteria, such as Bacteroides, increases. The abundance of Firmicutes, on the other hand, decreases [5,6]. Geographical factors also play a role in the composition of the microbiome [6]. The composition of the microbiome is affected by many drugs. Antibiotics have the best-known influence [7]. Other drugs include statins, metformin and proton pump inhibitors [8]. A significant effect is also seen with the use of probiotics and prebiotics [9], while a Mediterranean diet may have a possible impact [10]. Faecal transplantation, currently used, for example, in *C. difficile* infection, has in early studies shown the potential to improve cognitive function in AD patients [11,12]. Thus, the potential effects of GIT (gastrointestinal tract) traits on cognition and their underlying mechanisms, remain poorly understood. Moreover, available evidence on this subject comes from conventional observational studies (small sample sizes, confounding influences such as lifestyle), which may explain their inconsistent results [13].

There is also a possible risk of developing Alzheimer’s disease from drugs used for disorders of the gastrointestinal tract. Scopolamine, a drug used among others as an antispasmodic agent, has been used to induce AD in animal model studies. Scopolamine easily penetrates the blood–brain barrier. Its effects include increasing amyloid-β deposition and causing cholinergic dysfunction, which plays a significant role in the development of AD [14].

The epidemiology of the correlation between AD and GIT disorders is a subject of current research. It is estimated that there is an 11% increase in the risk of AD associated with exposure to *H. pylori* in patients at least 50 years old. There is no major effect modification by age or sex. Among 4,262,092 dementia-free subjects, 40,455 developed AD after a mean 11 years of follow-up [15].

For these reasons, research into the impact of gastrointestinal disorders on CNS diseases, including Alzheimer’s disease, requires further work. The purpose of this article is to provide an up-to-date overview of the correlation between Alzheimer’s disease and the gastrointestinal tract. Bibliographic research was conducted in May 2024 and was limited to articles in English published in the last 6 years. Articles were identified using the PubMed search engine. To avoid exclusion of important studies, studies were not restricted by type of publication or study design. The articles were selected on the basis of the corresponding abstract. Some publications were selected from the content of the searched publications. Flow diagram for the identification of selected studies is presented in Figure 1.

## 2. Selected Gastrointestinal Tract Disorders

### 2.1. Helicobacter pylori

*Helicobacter pylori* is a Gram-negative bacterium that is considered one of the human carcinogens. It is associated with gastric mucosal diseases, but studies have also revealed the involvement of *H. pylori* in the pathogenesis of extra-intestinal diseases, including neurological diseases. It is now suggested that there are multiple pathogenic mechanisms for this bacterium, including systemic inflammation of a minor degree, molecular mimicry and intestinal dysbiosis. The pathogen has multiple virulence factors, such as VacA and the CagA oncoprotein, which cause stimulation of immune pathways resulting in tumorigenesis and inflammation [16]. VacA causes vacuolisation and apoptosis of epithelial cells [17]. *H. pylori* infection is a global problem, the exact prevalence being linked to the socio-economic status of the population, with over 80% of adults infected in developing countries and 20–50% in industrialised countries [18].

AD is a neurodegenerative disease characterised by cognitive impairment and develops as a result of beta-amyloid accumulation in neurons, leading to their apoptosis. *H. pylori* and its harmful metabolites can enter the CNS via the oropharyngeal olfactory pathway, the retrograde gastrointestinal nerve pathway, infection of blood monocytes and disruption of the blood–brain barrier. Olfactory system dysfunction and degeneration are the earliest signs of AD and affect approximately 90 per cent of patients [16,18,19].

In a population-based, nested case-control study, exposure to clinically apparent *Helicobacter pylori* infection (CAHPI) was associated with a moderately increased risk of AD (OR, 1.11; 95% CI, 1.01–1.21). The increase in the risk was 7.3 to 10.8 years after CAHPI onset (OR, 1.24; 95% CI, 1.05–1.47). The study found an 11% increase in the risk of AD associated with exposure to CAHPI among subjects aged at least 50 years. The increase in the risk of AD reached a peak of 24% a decade after the onset of CAHPI [15]. 

Amyloid accumulation results from proteolytic processing of the APP protein. *H. pylori* infection may increase APP expression and contribute to CNS deposition of deposits. ApoE is the flagship cholesterol transporter in the brain and consists of three major allelic variants ε2, ε3, ε4, with ApoE4 contributing to higher AD risk and ApoE2 contributing to lower disease risk compared to the common ApoE3 phenotype. In studies, it has been shown that during *H. pylori* infection, there is an increase in the expression level of ApoE4 in cells, while there is a decrease in ApoE2. By altering ApoE levels, the pathogen may affect neurodegeneration, disrupting microglia, synapses, and lipid and glucose homeostasis [16]. BIN1, Clu, ABCA7 and CD33 are genetic susceptibility loci for AD, and *H. pylori* infection results in increased expression of neurodegeneration risk genes. The impact of TRL-4 receptor activation in AD pathogenesis has been demonstrated. This receptor recognises molecular patterns of DAMPs and PAMPs, such as HMGB1 or LPS of the bacterium *H. pylori*, which triggers an inflammatory response in the CNS. The pathogen has been shown to cause a significant increase in TRL-4 receptor expression in infected cells [16]. Furthermore, it has been noted that *H. pylori* causes an increase in the expression of pro-inflammatory cytokines, in particular CRP, IL-8 and TNF-α [16]. Infection has been shown to lead to blood–brain barrier (BBB) damage. TNF-α is involved in BBB disruption through a mechanism involving matrix metalloproteinase up-regulation. TNF-α and IL-6 play an important role in regulating the synthesis of other acute phase proteins that are risk factors for MetS-associated atherosclerosis, such as fibrinogen and factor VIII. These cytokines also affect lipid metabolism, which influences atherosclerosis, BBB damage and, indirectly, neurodegeneration. Activated monocytes enter the CNS from the blood and cause neuronal inflammation; in addition, pathogens, e.g., CMV or Chlamydia, also penetrate the damaged barrier [19]. *H. pylori* induces granulocyte infiltration, activation, maturation and antigen presentation. Antibodies against HP can cross-react with host antigens and damage them as a result of molecular mimicry between HP and human proteins. Among other things, neurons have homologous epitopes, so that antibodies react to their components, and this accelerates CNS damage [17,19]. In addition, HP has been shown to cause inflammation of the nervous system in mice by producing OMs containing bacterial components, including virulence factors. HP-OMVs can enter the blood through gastritis and are then transported through the damaged BBB to the brain, where they cause activation of microglia and astrocytes. The activated microglia produce inflammatory and toxic products that are involved in the accumulation of amyloid-β [17]. Significantly higher levels of specific anti-*H. pylori* IgG antibodies in CSF and serum are found in patients with AD compared to the general population. When AD and infection coexist, disease symptoms are more severe than in those without antibodies [18]. Galectins are lectin superfamily members, a group of endogenous glycan-binding proteins, able to interact with glycosylated receptors expressed by several immune cells. One of them, Galectin-3, recognizes β-galectocide molecules expressed by numerous bacteria, including HP, playing a role in innate immunity recruitment, and mediates activation of macrophages and neutrophils. During HP infection, gal-3 expression is up-regulated, resulting in disruption of the BBB and facilitating the entry of HP epitopes and immune cells into the CNS. In addition, gal-3 induces IL-8 production by lymphocytes, and this cytokine is responsible for promoting beta-amyloid deposition in the brain. Overproduction of gal-3 impairs the phagocytic capacity of macrophages, so that neuronal regeneration is reduced [20].

Drugs used in eradication therapy (“triple therapy”) are proton pump inhibitors, clarithromycin, and amoxicillin or metronidazole. Failure to respond to this treatment is likely due to resistance against antibiotics, especially in elderly people—it is estimated that these patients are 2–3 times more prone to antibiotic resistance. Geriatric patients take many drugs, and they also fail to respond to this treatment due to alteration in the interaction of the drugs, such as omeprazole, with cardiovascular medications [21].

Moreover, triple therapy may trigger neuropsychiatric symptoms and acute infection. The data suggest that psychiatric symptoms (dissociation, anxiety, mania, delirium, psychosis) usually disappear after discontinuing the antibiotics [18].

From the above information, it is likely that *H. pylori* infection may predispose to the onset and also progression of AD, but this thesis requires further confirmation in preclinical and clinical studies. Some of the pathogenetic factors are presented in Figure 2.

Researchers also investigated the association of proton pump inhibitor use with the risk of dementia. Consistent use of PPIs appeared to protect against cognitive decline among APoe4 carriers, while discontinuation of PPIs was more protective against cognitive decline among non-carriers. The temporary effect of PPIs on acetylcholine may have differential effects on cognitive flexibility among APoe4 carriers and non-carriers. In particular, acetylcholine is thought to play an important role in attentional processes associated with cognitive flexibility. For example, two studies have shown that acetylcholinesterase inhibition is associated with reduced impairment of attentional switching and may increase cognitive flexibility in rodents. However, the current study found no evidence to support the claim that PPI use is associated with an increased risk of cognitive decline associated with Alzheimer’s disease. However, the results suggest that PPIs may have different effects on changes in cognitive flexibility in APoe4 carriers and non-carriers [22].

Moreover, besides *Helicobacter pylori* infection, there are more infectious agents associated with Alzheimer’s disease, such as herpes simplex virus 1, HHV6, HHV7, *Borrelia burgdorferi*, *Chlamydia pneumoniae* and *Porphyromonas gingivalis*. In addition, some studies suggest that cytomegalovirus (CMV) and varicella-zoster virus (VZV) may also be associated with AD pathogenesis and further research should be performed [23,24].

### 2.2. Periodontitis and Oral Microbiome

Periodontitis is a chronic gum disease caused by an immune response to Gram-positive and primarily Gram-negative anaerobic bacteria, such as *Treponema denticola*, *Aggregatibacter actinomycetemcomitans*, *Porphyromonas gingivalis* and *Tannerella forsythia* [2]. Periodontitis can cause systemic inflammation of low severity that contributes to diseases related to other systems. Compared to healthy controls, patients with periodontitis present with high levels of pro-inflammatory substances (IL-1, IL-6, CRP, fibrinogen) and an increased number of neutrophils in the blood. Inflammation of areas of the body outside the oral cavity is also caused by translocation of dental bacteria [25].

The search for modifiable risk factors for the development of AD is ongoing, and research indicates that there is a link between the disease and chronic periodontitis and oral inflammation. The researchers emphasise that the link is bidirectional—people with periodontitis have an increased risk of developing dementia, and AD sufferers neglect oral hygiene due to cognitive decline (this can result in tooth loss and mucosal lesions). Periodontitis is associated with an approximately 1.7-fold increased risk of developing neurodegeneration. Bacteria (Gram-positive and then Gram-negative) that cause periodontal inflammation can penetrate through the epithelium of the periodontal pocket (periodontal pocket) into the bloodstream by brushing teeth, flossing or chewing food. They produce endotoxins and exotoxins that are transported via the bloodstream to, among other places, the brain, where they induce CNS inflammation, which is thought to be one of the possible causes of Alzheimer’s disease [26,27]. It has been shown that lipopolysaccharide, LPS (a component of the membrane of Gram-negative bacteria), can be found in high amounts in the brain of AD patients compared to healthy individuals. It has been shown that LPS accompanies beta-amyloid in plaques that form in the brain; in addition, it is also present around the brain vessels of patients. LPS activates microglia and induces inflammation (IL-1β, IL-6, IL-8, TNF-α and CRP) of the CNS in mouse models and can also increase the conversion of APP protein to insoluble deposits. This glycoprotein has the ability to induce the release of nitrous oxide and prostaglandin E2 from the glia. LPS activates cathepsin B-dependent inflammatory processes that promote the production of IL-1β and β-amyloid. Cathepsin B has been identified as a possible therapeutic target of AD [26,28]. Patients with AD and periodontitis had elevated serum levels of TNF-α and antibodies to *P. gingivalis*, A. actinomycetemcomitans and *T. forsythia*. *P. gingivalis* is a pathogen that can migrate to the brain and increase the production of beta-amyloid and tau protein [26]. In addition, *P. gingivalis* in mouse brains produces gingipains (proteolytic enzymes that degrade host proteins and activate inflammatory cells). The apolipoprotein E gene (ApoE, particularly the E4 allele) is a risk factor for AD because ApoE is involved in beta-amyloid metabolism. Fragmentation of ApoE causes neurodegeneration, and *P. gingivalis* gingipains leave ApoE, which may contribute to the formation of insoluble plaques in AD. These enzymes were identified in more than 90% of brains affected by AD taken post-mortem and were most prominent in areas related to memory (hippocampus). It is presumed that inhibition of gingipain activity may slow the progression of AD, so human clinical trials of drugs with this profile of action are currently underway [25,27]. *P. gingivalis* DNA has been detected in brain autopsy samples from AD patients and in the cerebrospinal fluid of living patients [25]. In addition, periodontitis causes a transient bacteraemia and, therefore, a systemic inflammatory response, which results in damage to the integrity and increased permeability of the blood–brain barrier (BBB). The consequence of this process can be synapse dysfunction and neuronal death in Alzheimer’s disease [2].

When considering the microbiome, it is important to remember that it does not only involve the gastrointestinal tract. A bidirectional gut–brain axis acting through the vagus nerve, tryptophan metabolism, neurotransmitters, the immune system, the hypothalamic–adrenal–pituitary axis, short-chain fatty acids and microbial metabolites ensures that the body’s homeostasis is maintained [29]. However, the human body is also colonised in places such as the skin or vagina in women and, importantly in the light of Alzheimer’s disease, the oral cavity. The gut microbiome shows similarities to the oral flora both in terms of its composition and diversity. Changes in the gut microbiota can be influenced by lifestyle or environmental factors, but pathogens also play a major role. The microbiota of the healthy oral cavity includes species such as *Gemella*, *Streptococcus*, *Granulicatella* and *Veillonella*, with *Streptococcus mitis* being found most commonly in each area. Their predominance in the oral cavity derives from their role in the formation of a biofilm on the surface of teeth, referred to as plaque. While it is often viewed as harmful, it actually plays an important protective and regulatory role, provided the microbiome is in balance. Plaque acts as a barrier against external microorganisms and, therefore, plays a beneficial role. The case is different when this equilibrium is disrupted—whether the biofilm is depleted or overgrown. Any such disruption can promote colonisation of the oral cavity by pathogenic organisms, leading to oral infections. This allows the growth of *Porphyromonas gingivalis*, which can normally be found in the oral cavity in trace amounts, leading to periodontitis [30]. Patients with periodontitis can ingest up to 10^12^ bacteria per day along with their saliva. Under normal conditions in a healthy person, commensals such as *Streptococcus* and *Actinomyces* exhibit a physiological symbiosis and do not lead to disease development. However, any disruption to the natural state of the oral cavity can lead to the emergence of a dysbiotic microbial community, which includes *Firmicutes*, *Spirochaetes*, *Bacteroidetes* and *Proteobacteria*. Playing a role in numerous systemic diseases, such as rheumatoid arthritis, cardiovascular disease and diabetes, they contribute to the systemic inflammatory response and thus neuroinflammation and AD [31]. Periodontal pathogens, such as *Aggregatibacter actinomycetemcomitans*, *Prevotella gingivalis* and *Tannerella forsythia*, have also been detected in AD patients [32]. Periodontitis should be considered as a modifiable risk factor for AD due to the presence of bacterial molecules such as flagellin, peptidiglycanor LPS and pro-inflammatory molecules. Oral inflammation may consequently lead to an imbalance of the intestinal microflora and exacerbate the systemic immune response. Oral administration of *P. gingivalis* in mice, mimicking the effects of periodontitis, altered their gut microbiome, caused dendotoxaemia and increased macrophage infiltration into adipose tissue, as well as an inflammatory response in the liver by increasing the pro-inflammatory cytokines TNF-α, IL-6, Fitm2 and Plin2, which, in turn, may indicate a link between oral infection and bacterial ingestion and transfer of the gut microbiota to the liver. In addition, the study revealed significant cognitive impairment and greater deposition of Aβ deposits, as well as higher levels of pro-inflammatory cytokines, such as IL-1β and TNF-α, in their brains, compared to control mice. Gingipain was detected in the brain tissue of the mice, in both neurons and glial cells, confirming the translocation of bacteria from the oral cavity to the brain. The study also resulted in changes in the proportions of bacteria observed in mouse faeces. The proportions of *Bacteroidetes* appeared to be significantly lower, while *Firmicutes* increased noticeably. The changes in intestinal composition resulted in Th17 dominance. Studies have also shown that periodontitis can lead to chronic low-grade systemic inflammation, which, in turn, can act as an inducer or exacerbator of neuroinflammation, ultimately leading to neurodegenerative changes and Alzheimer’s disease [31,33]. Other studies have also shown learning deficits and memory impairment in rats with periodontitis and significant memory loss in older individuals, compared to younger individuals from control groups. A role for inflammation in AD was also demonstrated by linking Aβ to the complement system, whose activated components were linked to deposits [33]. The doubled risk of Alzheimer’s disease concerning periodontitis is also associated with tooth loss in its course. A similar trend has been observed with irregular tooth brushing. Tooth loss can lead to chewing disorders, which, affecting nutritional status, results in reduced cerebral blood flow, associated with memory deficits [34]. To investigate the impact of oral homeostasis and microbiota on the development of sporadic AD and dementia, we looked at ten years of dental records from Taiwan’s National Insurance database. It showed that occasional, single dental procedures, such as tooth extractions or root canal treatment, tended to reduce patients’ risk of developing dementia. However, when the need for procedures increased beyond four, the risk of AD increased. The above trend, therefore, points to chronicity of the disease as a key concern. Taking care of oral hygiene and maintaining oral homeostasis by remaining under dental care was associated with a lower risk of AD, which may support the hypothesis that the oral microflora is a factor in the development of sporadic AD [35].

In an attempt to understand the pathogenesis of sporadic Alzheimer’s disease, a number of potential microorganisms have been isolated in autopsy studies of the brain, which in model experiments have shown the ability to accumulate Aβ deposits and neurofibrillary degeneration consisting of tau protein. Most of them cope with host defence mechanisms and are able to survive through latency in internal reservoirs, resulting in their periodic reactivation and promoting a persistent and chronic inflammatory process. Despite the initial proposal of major causative agents, the role of a single pathogen is unlikely, and multiple microorganisms have been routinely isolated from brain samples of AD patients. The entry of one pathogen into the brain appears to be a factor that opens the door for further pathogens, further increasing their virulence. It also stimulates a stronger immune response in the body. Many microorganisms isolated from the neural tissue of people with sporadic AD have been found in the gingival region, from where they enter the brain through rich vascularisation and retrograde transport along the cranial nerves [35].

### 2.3. Inflammatory Bowel Diseases

Inflammatory bowel disease (IBD) results from the interaction between genetic and environmental factors which influence the immune responses. IBD is mainly divided into ulcerative colitis (UC) and Crohn’s disease (CD). The immune system is stimulated, and intestines are destroyed. Symptoms include diarrhoea, abdominal pain, rectal bleeding, weight loss (malnutrition), and behavioural manifestations, including anxiety, depression and cognitive dysfunctions. IBD develops in young adulthood and continues throughout life. Both the diseases may affect men and women equally [36]. Emerging evidence from observational studies indicated that IBD may be associated with risk of dementia. Existing meta-analyses demonstrate a positive unidirectional relationship between IBD and dementia, with most studies reporting the hazard ratio to be greater than 1 and less than 2. However, the findings are not consistent, and some studies do not support an association between IBD and AD. Moreover, IBD therapies, such as immunosuppressants and TNF-α blockers, may play an important role in AD development by controlling inflammation, which may result in a lower risk of AD in these individuals [37,38].

Several plausible mechanisms link IBD with neurodegeneration, including genetic factors, gut microbiome dysbiosis and environmental factors. It was shown that patients with IBD were diagnosed with dementia earlier compared to matched controls (mean age 76 versus 83) and risk was elevated with increased chronicity of IBD [38]. Increased intestinal permeability in IBD leads to neuronal inflammation from transference of intestinal inflammation. One potential approach is to use the vagus nerve, which serves as a link between the ENS and CNS. Chronic inflammation in IBD may lead to beta-amyloid plaque formation in the intestinal tract that spreads to the brain via the vagus nerve. Following damage to the intestinal epithelium, bacteria and their metabolites translocate to the intestinal wall, where immune cells become activated through TLR signalling and secrete a large number of pro-inflammatory cytokines—IL-1β, IL-6, IL-18 and TNF-α. TNF-α is the main pro-inflammatory cytokine in IBD pathology, and in people with Alzheimer’s disease, increased serum levels of TNF-α exacerbate cognitive impairment and neurodegeneration through activation of microglia [2,38]. Another link between IBD and dementia is dysbiosis of the gut microbiota. It has been found that both IBD and AD patients have a reduced diversity of the microbiome, which in healthy individuals produces the anti-inflammatory substances SCFAs, certain bile acids, including tauroursodeoxycholic acid, and aryl hydrocarbon receptor ligands, which are capable of crossing the BBB to alleviate inflammation in the CNS. Decreased production of these beneficial metabolites is secondary to intestinal dysbiosis. In addition, several studies have also noted that chronic inflammation in IBD is accompanied by the production of neurotoxic metabolites (this includes cinurenin, LPS, enterotoxins), which, in turn, promote inflammation in the CNS by activating microglia and astrocytes. These substances damage the intestinal lining, resulting in an increased ability of neurotoxic metabolites to migrate from the intestinal lumen into the circulation and then into the CNS [38]. However, current evidence does not support an association of ulcerative colitis and Crohn’s disease with the risk of Alzheimer’s disease. Furthermore, a study by Adewuyi EO and colleagues found a lack of genetic correlation between IBD and AD, which was different from the correlation in the other gastrointestinal diseases considered in the study [13]. Because of the different conclusions of the various studies, there is still a need for new research on this topic.

### 2.4. Adiponectin and Obesity

Obesity has been consistently associated with Alzheimer’s disease, though the exact mechanisms by which it influences cognition are still elusive and the subject of current research. Adipose tissue is considered as an endocrine organ, capable of secreting factors that regulate physiological functions. Adiponectin (ADPN) is one such factor, secreted in large quantities and which suppresses glucose production in the liver, enhances fatty acid oxidation in skeletal muscle, protects cells from apoptosis and reduces inflammation in cells via receptor-dependent mechanisms. The first important role of adiponectin in Alzheimer’s disease is that it is considered as an immune mediator. Receptors (AdipoRs) occur on the surface of immune cells and adiponectin modulates their function, response, proliferation and polarisation. AdipoR1 activation induces the suppression of proliferation of M1 macrophages and the expression of pro-inflammatory cytokines. AdipoR2 activation modulates the polarisation of anti-inflammatory M2 macrophages. Reduction in adiponectin serum levels correlates inversely with the severity of the neurodegeneration. Most studies showed the presence of AdipoR1 in the hypothalamus and in the Meynert basal nucleus, suggesting a feasible adiponectin involvement in neuroprotective function. AD pathogenesis is characterised by deregulation of the brain AMPK pathway, which, in turn, could phosphorylate tau protein, causing changes in the brain. Through AMPK, adiponectin increases neuronal insulin sensitivity by p-Akt. Conversely, chronic adiponectin deficit inactivates AMPK, reducing neuronal insulin sensitivity, and inducing AD in elderly mice [39].

An anti-apoptotic role is also assigned to ADPN and carried out by the activation of the enzyme ceramidase and consequently the enhancement of its metabolite, sphingosine-1-phosphate (S1p), which is involved in survival pathways. It is a potent neuroprotective factor against soluble Aβ-induced apoptosis and promotor of long-term potentiation, which is essential to memory consolidation. In the hippocampus and temporal cortex, decreased levels of S1p have been associated with a higher neurofibrillary tangle (NFT) burden. ApoE regulates the secretion of S1p, and the hippocampal S1p/sphingosine ratio is higher in ApoΕe2 carriers compared to ApoEe4 carriers, linking this sphingolipid to the most relevant genetic risk factor for late onset AD. Thus, S1p constitutes one more possible convergent molecule in ADPN and AD metabolisms. Moreover, a couple of studies have reported diminished hippocampal neurogenesis in ADPN-haploinsufficient and/or ADPN-deficient mice, which are reversed with intracerebroventricular (ICV) ADPN administration. Chronic ADPN deficiency in aged ADPN-knockout mice leads to AD-like pathology and cognitive deficits, reinforcing the relevance of ADPN in this disease [40].

### 2.5. Microbiome

#### 2.5.1. General Information

The mammalian gut is inhabited by up to 1000 different species of bacteria. The four dominant bacterial clusters are *Firmicutes*, *Bacteroidetes*, *Actinobacteria* and *Proteobacteria*. Under physiological conditions, the mucosal microbiota plays an important role in food digestion, angiogenesis, vitamin synthesis, maturation, development and education of the host immune system [41]. The gut microbiota in a healthy body influences pH and forms a barrier against infectious agents [23]. Taking antibiotics, poor eating habits, excessive alcohol consumption, smoking and stress weaken the actions of beneficial gut bacteria or alter their composition, leading to an imbalance and compromising health [23]. The composition of the microbiome differs significantly between healthy people and those affected by neurodegenerative diseases, for example, Alzheimer’s disease (AD), Parkinson’s disease and multiple sclerosis [41]. Other diseases associated with negative changes in the microbiome include metabolic syndrome, obesity, inflammatory bowel disease, colorectal cancer, type 2 diabetes and heart failure [23].

#### 2.5.2. Microbiota–Gut–Brain (MGB) Axis

There is a strong bidirectional relationship between the gut and the central gastrointestinal system. The gut–brain axis, otherwise known as the microbiota–gut–brain (MGB) axis, integrates peripheral gut function with the emotional functions and cognitive centres of the brain via neural, hormonal, immune and metabolic pathways [42,43]. The vagus nerve connects the gut to the brainstem. Its nuclei in the trunk send signals to different areas of the brain, including the cortex and thalamus [23]. Gut bacteria are receptive to neurotransmitters sent by the brain. They also produce substances, such as monoamines and amino acids, which reach central neurons via the lymphatic system and affect their activity [23]. Metabolites produced by bacteria can positively or negatively affect brain function. Short-chain fatty acids (SCFAs), consisting mainly of butyrate, propionate and acetate [10], play a role in maintaining homeostasis in the CNS. Based on rodent studies, it appears that SCFAs mainly affect the hippocampus and striatum, which may translate into significant modelling of cognitive functions, including learning, as well as reward-related behaviour. SCFAs also reduce anxiety and depressive behaviour [8]. Short-chain fatty acids have been shown to be effective in disrupting protein–protein interactions, which are essential for Aβ assembly. They are also active mediators of gut–brain communication [10]. Another metabolite produced by bacteria has been linked to the pathogenesis of Alzheimer’s disease. Trimethylamine N-oxide (TMAO) increases β-secretase activity and thereby enhances Aβ accumulation, which triggers pathological processes typical of AD. By enhancing the stimulus-dependent release of calcium ions from intracellular stores, TMAO induces platelet hyperreactivity. The production of Aβ in platelets and its passage from the bloodstream to the brain is increased [10]. SCFAs and TMAO also affect changes in blood–brain barrier (BBB) permeability. SCFAs can modulate the expression of tight junction proteins, while TMAO alters the expression of annexin A1 protein, which increases blood–brain barrier integrity. TMAO also reduces astrocyte- and microglia-dependent nervous system inflammation, which has the effect of reducing lipopolysaccharide-dependent memory loss [11]. The BBB in elderly patients shows increased permeability, which is associated with a risk of being overrun by pathogenic microorganisms. Bacteria in the brain induce an immune response, which by multiple mechanisms leads to the activation of microglia. Activated microglia produce inflammatory mediators, such as IL-1β, IL-6, TNF-α, iNOS and reactive oxygen species (ROS), which induce necrosis and apoptosis of dopaminergic neurons in the CNS. Inflammation of the nervous system has been implicated in the development of Alzheimer’s disease. Some bacteria can also negatively affect the permeability of the blood–brain barrier. Lipopolysaccharides (LPS) of the bacterium *Fusobacterium nucleatum* have been shown to increase its permeability [44]. Neurons release substances that sustain the inflammatory process and immune response, which is referred to as ‘neuroinflammation’. This can be a beneficial or detrimental process for the brain, depending on the strength of the activation [23]. It has been shown that a prolonged neuroinflammatory process can cause some neurodegenerative diseases, including Alzheimer’s disease. These patients, in particular, have been shown to have increased serum levels of pro-inflammatory cytokines, such as interleukin-1, interleukin-6, TNF-alpha and TGF-beta [23]. Alzheimer’s disease is characterised by systemic as well as intestinal inflammation [45]. Substances with very potent pro-inflammatory effects include lipopolysaccharides, or glycolipids derived from the microbiome. In humans, facultatively anaerobic gastrointestinal Gram-negative bacilli, including *Bacteroides fragilis* and *Escherichia coli*, are the main source of LPS [46]. The microbiota–gut–brain axis is presented in Figure 3.

#### 2.5.3. Composition and Role of the Microbiome in Alzheimer’s Disease

The composition and abundance of bacteria present in the gut microflora affect brain function. The composition of the microflora changes with age. In younger people, *Firmicutes* are present in abundance far above *Bacteroides*. This ratio is reversed with age. In older individuals, the diversity of the microflora decreases with a predominance of pro-inflammatory bacteria [6]. In symptomatic AD, a dysbiosis of the microflora relative to healthy individuals is evident [5]. In AD, a decrease in the diversity of the microbiome with a predominance of pro-inflammatory species is typical [12]. There is an increased relative abundance of *Bacteroidetes* and a decreased relative abundance of *Firmicutes* [5], including the pro-inflammatory Faecalibacterium prausnitzii. Increased abundance of Bacteroides increases the translocation of LPS from the gut into the systemic circulation, leading to inflammation [6]. Other changes in the microbiome found in Alzheimer’s patients include increased abundance of Desulfovibrio parabiont and decreased abundance of Clostridium sensu stricto 1, which has been linked to adverse effects of AD [45]. Additionally, the presence of *Eubacterium rectale*, *Porphyromonasgingivalis* and *Lactobacillus rhamnosus* may play an important role in the pathogenesis of AD [43]. It is also worth noting that the composition of the gut microbiome depends on ethnic and geographical factors. When comparing studies conducted in the USA and China, differences are noticeable. In the case of studies from the USA, an increase in the abundance of Bacteroides and Acinetobacter was noted with a decrease in the volume of Bifidobacterium; in Chinese studies the results were the opposite [6]. In a mouse model, an increase in Lactobacillus casei, Bacteroides fragilis and Streptococcus thermophilus showed positive effects on brain activity and performance [43]. A summary of the bacteria for which abundance increases in AD and which have been linked to an increased risk of developing Alzheimer’s disease and those for which abundance decreases in AD is shown in Table 1.

A study in mice showed a significant correlation between a dynamic change in the composition of the gut microflora during AD progression and an increase in Th1 cell infiltration. The number of Th1 cells may be increased by, among other things, an abnormal, excessive production of isoleucine and phenylalanine by the gut microflora. These amino acids are involved in promoting both differentiation and proliferation and the passage of inflammatory Th1 cells from the peripheral circulation to the brain. Th1 cells activate M1 microglia, leading to microglia differentiation towards a pro-inflammatory state and increasing nervous system inflammation [47]. In studies in a mouse model, the complex gut microbiome has been shown to be one of the drivers of behavioural abnormalities, microglia activation and AD-specific changes. Additionally, the microbiota taken from older 3xTg mice accelerates the development of AD in 3xTg young mice accompanied by active C/EBPβ/AEP signalling. This indicates that changes in the microbiome with age influence AD pathology [48]. In another study, the microbiota of AD patients was transplanted into young rats. This induced the rodents to develop basal AD symptoms and impaired adult hippocampal neurogenesis (ANH). ANH is the process of creating new neurons from stem cells, and is a key mediator of many cognitive functions, including pattern separation and spatial learning, as well as emotion regulation. It has been suggested that ANH impairment is an early feature of AD pathogenesis [45]. Partners of AD patients have been shown to have an increased risk of developing a spontaneous form of dementia. The oral and intestinal microflora of the partners of AD patients differed from a healthy control group and resembled that of AD patients, indicating microflora transmission. Similar results have been obtained in animal studies, for example, in the co-culture of wild mouse and AD transgenic mice [49]. The potential role of gut dysbiosis in neurodegeneration and Alzheimer’s disease is shown in Figure 4.

#### 2.5.4. The Role of the Microbiome in Preclinical Alzheimer’s Disease

Studies suggest that there is a minimum of 10 years between the first deposition of Aβ plaques in the brain and AD symptoms. Preclinical AD is a condition in which cognitive function remains normal, with the presence of biomarkers indicative of disease, such as Aβ plaques detected by PiB or 18F-florbetapir (AV45) radioligand during PET imaging and CSF assays of Aβ42, Aβ40 and tau [5]. A study of patients classified as preclinical AD showed a different microbiome composition and functional potential of microorganisms. Microbiome pathways associated with preclinical AD in regression models include succinate as a product. Succinate is recognised as an immunomodulatory agent and is a major precursor of short-chain fatty acid propionate [5]. Bacterial taxa associated with both preclinical AD in the regression model and symptomatic AD are Alistipes, Barnesiella and Odoribacter. Different Bacteroides species are typical of healthy individuals (Bacteroides caccae) and individuals with preclinical AD (Bacteroides intestinalis). In symptomatic AD, changes in the composition of the microflora are noted compared to healthy individuals; there is an increased relative abundance of Bacteroidetes and a decreased relative abundance of Firmicutes. Such a ratio was not observed in a comparison of one study of patients with preclinical AD to healthy individuals [5]. In another study, however, there was evidence of a decrease in the abundance of Firmicutes and Bifidobacteria and an increase in the abundance of Bacteroides already in the pre-symptomatic stage [6].

#### 2.5.5. Role of Drugs and Diet

The best-known group of drugs affecting the volume and composition of the gut microflora are antibiotics. Their effect on the intestinal microflora has been known for a long time; however, their association with the development and progression of AD is not yet fully proven. The possible effect of use varies depending on the antibiotic; amoxicillin, rapamycin, rifampicin, doxycycline and minocycline improved cognitive function, and affected reduced accumulation of hyperphosphorylated tau and Aβ decay, while streptozotocin, ampicillin and cefepime were associated with memory deficits, confusion and reduced consciousness. Additionally, streptozotocin in animal models has shown the ability to induce sporadic AD [7]. Other drug groups with significant effects on the gut microflora include metformin, proton pump inhibitors (PPIs), statins and laxatives. Drugs with strong effects on the abundance of individual species include antidepressants, antipsychotics and PPIs. For example, serotonin reuptake inhibitors, including sertaline, fluoxetine and paroxetine, show antimicrobial activity against Gram-positive bacteria, such as Staphylococcus and Enterococcus [8,50]. Selected medications and their impact on the microbiome, which has been studied in connection with the pathogenesis of Alzheimer’s disease are presented in Table 2.

Increasing attention is now focusing on the impact of diet, including the use of probiotics and prebiotics as a form of adjuvant therapy for AD. The most commonly used probiotics are lactic acid bacteria, especially *Lactobacilli*, *Enterococcus*, *Streptococci*, *Bifidobacteria*, *Pediococcus* and some yeasts, such as Saccharomyces boulardii [43]. Prebiotics are organic substances that are not digested in the human intestine and have the ability to stimulate the growth and activity of intestinal bacteria. For example, in an animal model, the use of yeast beta-glucans has been shown to have a positive effect in balancing between pro-inflammatory and anti-inflammatory intestinal bacterial species, thereby reducing nervous system inflammation and insulin resistance. A reduction in nervous system inflammation was also observed with lactulose and melibiose. A reduction in blood–brain barrier dysfunction was also observed in association with the use of mannan oligosaccharide [12]. Studies have shown that supplementation with Bifidobacterium breve A1 can induce positive effects on cognitive function in people with memory problems, reduce inflammation in the hippocampus and suppress the immune-reactive genes that are induced by amyloid. Several studies have demonstrated the positive effects of a balanced diet rich in prebiotics and probiotics, as well as other important nutrients, in delaying cognitive decline and reducing the risk of Alzheimer’s disease. Significant improvements in cognitive function following a diet high in fermented dairy products also occurred in patients already diagnosed with Alzheimer’s disease. Consumption of bioactive peptides contained in dairy products or tryptophan-related dipeptides and new lacto-peptides contained in fermented dairy products has a positive effect on cognitive function [9]. Also, a Mediterranean diet (MD), rich in fruit, vegetables and legumes, may be helpful in AD therapy. According to human intervention studies, such a diet can modulate the composition of the gut microbiome, increase faecal SCFA levels and decrease urinary TMAO levels. MD produces an anti-inflammatory effect, which is often associated with an increase in Bacteroides and Clostridium and a decrease in Proteobacteria and Bacillaceae [10].

#### 2.5.6. Other Therapeutic Options

Faecal transplantation (FMT), which restores the eubiosis of the intestinal microflora, is mentioned as a possible form of AD therapy. Faecal transplantation is already used as a therapy for other conditions, for example, C. difficile infection. In animal model studies, FMT has been shown to be effective in reducing cognitive impairment, reducing Aβ, decreasing tau protein hyperphosphorylation and, depending on the study, has produced other positive results, e.g., reducing pro-inflammatory markers, increasing synaptic plasticity, increasing SCFA production [11,12]. Examples of the use of faecal transplantation in adjuvant therapy for AD in humans are very limited. Improvements in cognitive function have been documented in the available studies [12]. In a recently completed phase 3 clinical trial of GV-971, which is a carbohydrate-based anti-AD drug, its efficacy in reversing cognitive impairment in patients with mild to moderate Alzheimer’s disease was demonstrated. It is indicated that the therapeutic effect of GV-971 lies primarily in the restoration of the gut microbiome. Over the course of the study, the drug regenerated the intestinal microflora, reduced faecal and blood phenylalanine and isoleucine concentrations, and reduced inflammation in the nervous system associated with Th1 infiltration [47]. 

#### 2.5.7. Gut Microbiome and Synaptic Dysfunction

A major problem in the pathogenesis of Alzheimer’s disease is the accumulation of Aβ deposits and hyperphosphorylation of the tau protein. Recognised by microglia cells and astrocytes, they are the cause of an immune response leading to neuroinflammation and the production of pro-inflammatory cytokines. However, another mechanism plays a role here, which is the disruption of synaptic transmission between neurons. By producing reactive oxygen and nitrogen species, Aβ deposits can affect neuronal membrane damage, causing changes in dendritic spines, which also plays a significant role in cognitive decline [51]. The gut microbiome refers to microorganisms including bacteria, viruses, protists, archaeons and fungi, and their diversity is host-specific and can be determined by both environmental and genetic factors. The influence of the microbiota is now indicated not only on neuroinflammation, but also on the immune stability of the brain influenced by the peripheral immune system. The immunomodulatory and neuromodulatory actions of the microbiota overlap. Studies have shown that manipulation of the microbiome with antibiotics induced significant changes in gene expression in both microglia and neurons of the medial prefrontal cortex, which affected the dendritic plasticity capacity of neurons in learning processes. These changes persisted even after vagotomy, which may indicate a direct effect of microbiota metabolites on neuronal activity [52]. The main excitatory neurotransmitter in the cerebral cortex is glutamic acid. Under physiological conditions, glutamate is released into the synaptic gap, which activates membrane AMPA receptors (AMPARs) on postsynaptic neurons and allows the formation of the postsynaptic potential (EPSP). Early stages of Alzheimer’s-like pathology are likely to be associated with increased glutamate release. Pro-inflammatory cytokines, such as TNF-α and IL-1β, produced by microglia in the presence of deposited Aβ deposits, may increase AMPAR expression at the postsynaptic membrane, resulting in greater depolarisation. This leads to neuronal hyperpolarisation in the hippocampus and increased long-term potentiation (LTP) in the early stages of AD. LTP is a persistent strengthening of synapses based on recent patterns of activity, underpinning synaptic plasticity responsible for learning and memory. Metabolites of the microbiome, such as short-chain fatty acids (SCFA), bile acids (BA) and trimethylamine N-oxide (TMAO), have been identified in the brain of AD patients. It has been postulated that TMAO has an effect on LTP impairment, which negatively affects synaptic plasticity. However, studies have been conducted on ex vivo hippocampus incubated in the presence of TMAO, so it is uncertain whether a direct effect on neurons or on microglia plays a role. The result of the paradoxical hyperreactivity of glutamatergic neurotransmission in the early stages is its reduced activity in the later stages—chronic stimulation of AMPA receptors leads to their desensitisation. Higher levels of Aβ deposits affect the reduced release of glutamate by presynaptic neurons. Excessive production by microglia of pro-inflammatory cytokines, including TNF-α and IL-1β, can lead to neuronal death and additional glutamate release, exacerbating neurotoxicity. As a result, LTP is reduced, manifesting as cognitive deficits [52]. The neurotransmitters, synapses and cytokines in Alzheimer’s disease progression are presented in Figure 5.

#### 2.5.8. Amyloid Beta Cascade Hypothesis and Gut Microbiome

A key factor in the development of Alzheimer’s disease is the accumulation of insoluble amyloid β deposits forming into senile plaques, a premise of the so-called ‘amyloid cascade hypothesis’. Post-mortem studies show that the brain area particularly affected in AD patients is the hippocampal formation, especially its regions, such as the cornuammonis 1 (CA1), subiculum and entorhinal cortex. The 5xFAD mouse model is widely used as a preclinical model of AD, which shows overexpression of the human APP (amyloid precursor protein) and the PSEN1 (presenilin-1) gene, leading to more rapid amyloid plaque formation. Using the given model, we looked at factors that may influence the development and course of AD, such as the genetic profile—homozygote/heterozygote— and the composition of the gut microbiome at early and late behavioural stages. The abundance of clusters such as *Firmicutes* and *Bacteroidetes*, an indicator of microbial balance in the gut, was compared and showed that the ratio of one cluster to the other decreased significantly with age [53]. Another study analysed reactive gliosis surrounding Aβ plaques using the brain microglia marker Iba1. Using anti-Iba1 and Aβ antibodies, immunofluorescence staining was used to reveal the distribution of microglia in relation to amyloid deposits in APP/PS1 mice. In young mice, the microglia were evenly distributed, while Aβ plaques were absent. In older mice, clear clustering of microglia around Aβ deposits was detected. APP/PS1 mice were compared to young control mice noting significant differences in microglia abundance between them. Given that there was no accumulation of Aβ plaques or activation of microglia in the control group, the suggestion was made that a change in the gut microbiome occurs prior to amyloid beta cascade activation in the brain of Alzheimer’s disease model mice [54]. PSEN1 and PSEN2 gene expression was also tested for the effects on gut microbiota. Induced deletion of these genes resulted in spontaneous development of intestinal inflammation in mice, disruption of the epithelial barrier and translocation of intestinal bacteria, which was not observed in the control group without genetic factors. At the molecular level, intestinal epithelial cells lacking PSEN showed impaired Notch signalling and impaired epithelial differentiation. Mutations in these genes, therefore, affects the homeostasis of the intestinal microbiome, linking the issue with Alzheimer’s disease [55].

#### 2.5.9. Role of Organelles in AD and the Microbiome

Among the many interactions of the gut microbiome in maintaining the body’s homeostasis, the relationship between the microbiota and mitochondria should be mentioned. These organelles produce the ATP necessary for the physiological functioning of the body, participate in the transmission of metabolic and stress signals, and are sensitive to the metabolites produced by microorganisms associated with the mucous membranes and gastrointestinal tract. With age, the composition of the microbiome loses its diversity and abundance, which translates into mitochondrial functionality and energy production in intestinal cells, which, by reducing the integrity of intercellular junctions, affects the greater permeability of bacterial products, such as LPS. This phenomenon promotes inflammation, which plays a role in Alzheimer’s disease [56]. Hormones also play a role in AD, through energy regulation and food intake. Ghrelin, a peptide neurotrophic hormone secreted in the body during starvation, exerts neuroprotective effects by affecting mitochondrial respiration. In Alzheimer’s disease, impairment of memory and learning is significantly associated with an age-related decrease in plasma ghrelin levels [56]. Another factor underlying the pathogenesis of AD is defective proteolysis, leading to the accumulation of harmful deposits. The ubiquitin-proteasome system (UPS) is the main intracellular system that removes misfolded proteins, and disruption of the proteasome impairs both APP processing and Aβ production. Lysosomal enzymes, such as cathepsin L and cathepsin B, can interfere with APP processing, which alters the formation of Aβ deposits. Aβ accumulation and removal remain tightly regulated by the UPS, and disruption of proteolysis promotes the accumulation of abnormal Aβ structures, which is typical of neurons in Alzheimer’s disease [56]. Apolipoprotein E, especially ApoE-ε4, can impair the mitochondrial structure, which reduces mitochondrial activity and causes free radical accumulation, resulting in oxidative stress. Loss of mitochondrial function affects APP expression and processing, resulting in Aβ accumulation. Furthermore, the interaction of Aβ with mitochondrial proteins, inner membrane and matrix impairs oxidative phosphorylation, increasing the synthesis of reactive oxygen species (ROS). High levels of oxidised bases, such as 5-hydroxyuracil and 8-hydroxyguanine, were observed in the frontal, temporal and parietal lobes of the brain, indicating an association of Alzheimer’s disease with mitochondrial and nuclear DNA oxidation. Similarly, the gut microbiome, via the metabolite trimethylamine N-oxide (TMAO), whose levels increase with age, can induce oxidative stress [57]. What is more, APOE4 levels may influence the gut microbiota patterns. Erysipelotrichia abundance is increased by Apolipoprotein E in both mice and humans compared to populations without mutations of those alleles [58]. 

#### 2.5.10. Microbiome Imbalance and Gene Expression

The set of genes of gut bacteria consists of approximately 2.23 × 10^7^ genes. In contrast to the negative role of the microbiome, the genus Bacteroides has the potential to release a significant amount of neurotoxins, the most common of which are lipopolysaccharides (LPS), truncated forms of LPS molecules known as lipooligosaccharides (LOS), related enterotoxins, small noncoding RNAs (sncRNAs) and bacterial amyloids. Extremely strong neurotoxins are secreted by the strain of *Bacteroides fragilis*—a glycolipid subtype BF-LPS. In recent years, LPS and BF-LPS have been shown to bind to the nucleus circuit of neurons in the brain of people with Alzheimer’s disease and to promote the transcription of the factor NF-kB in human neuronal-glial cells. BF-LPS derived from Gram-negative resident cells of the gastrointestinal tract are an important factor in down-regulating the specialised gene expression patterns required for normal CNS homeostasis, which translates into the progressive neurodegenerative changes observed in AD [59].

## 3. Conclusions

Considering the above issues, it can be seen that Alzheimer’s disease is a much broader concept than just a disease of the nervous system. In its pathogenesis, although still not fully understood, diseases related to the digestive system play a role, such as nonspecific intestinal inflammation or diseases related to the oral cavity, with periodontitis at the forefront. The widely understood homeostasis of the human body, based on the balance of the microbiome—both the one related to the intestines and the oral cavity—has a significant connection with the disease. Numerous observations have shown that disturbances in the amount or composition of the microbiome can affect the development of dementia, play a role in synaptic dysfunction and promote the development of sporadic Alzheimer’s disease. The gut–brain connection reveals the profound potential links to neurological disorders. Understanding this link holds the key to unlocking new therapeutic pathways in AD. Future research should focus on the impact of other gastroenterological diseases on the onset of Alzheimer’s disease. It is important to find further infectious agents that may be associated with this disease. Moreover, subsequent research should extend the topic related to connections between microbiome disorders and neurodegenerative diseases. 

## Figures and Tables

**Figure 1 nutrients-16-02366-f001:**
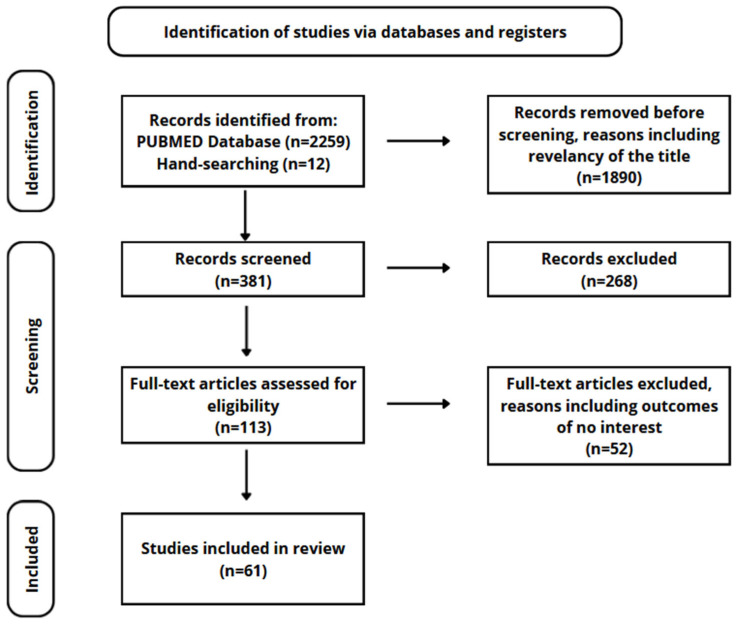
Flow diagram for the identification of selected studies.

**Figure 2 nutrients-16-02366-f002:**
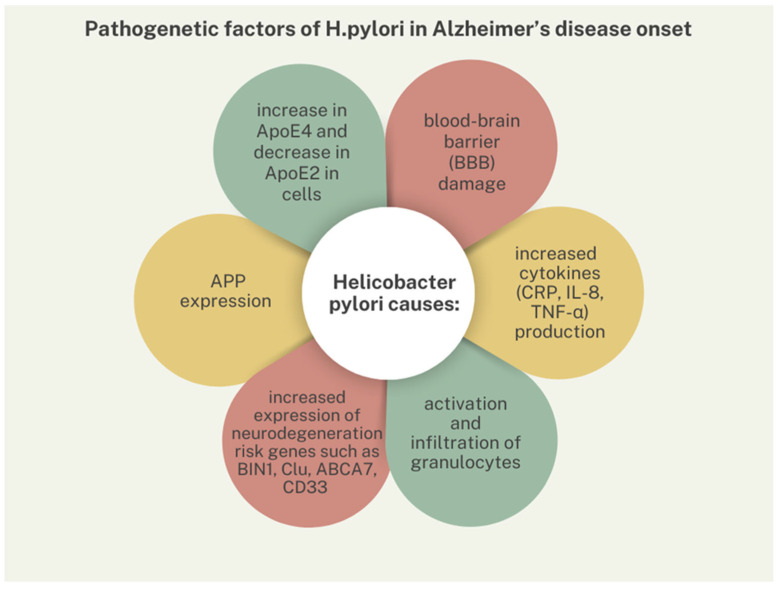
Pathogenetic factors of *H. pylori* in Alzheimer’s disease onset.

**Figure 3 nutrients-16-02366-f003:**
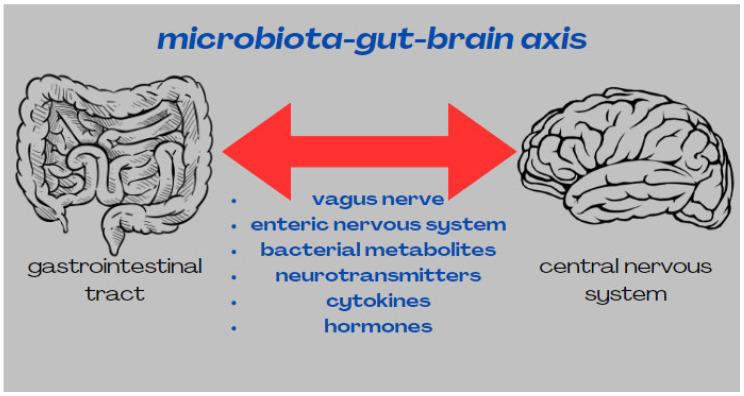
Microbiota–gut–brain axis.

**Figure 4 nutrients-16-02366-f004:**
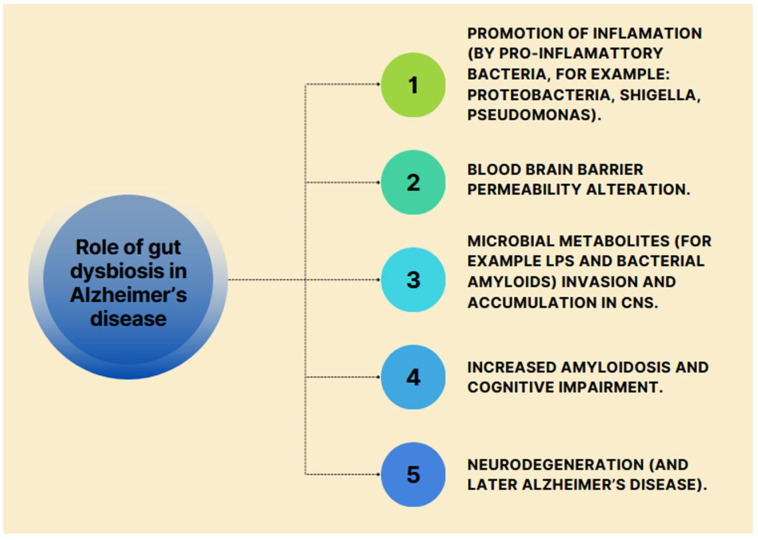
Role of gut dysbiosis in Alzheimer’s disease [29].

**Figure 5 nutrients-16-02366-f005:**
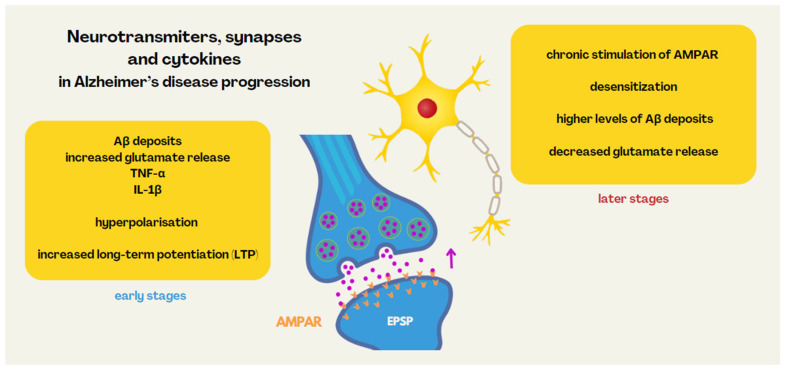
Neurotransmitters, synapses and cytokines in Alzheimer’s disease progression.

**Table 1 nutrients-16-02366-t001:** Examples of bacteria connected to Alzheimer’s disease.

Bacteria Linked to an Increased Risk of Developing AD (Bacteria for Which Abundance Increases in AD)	Bacteria for Which Abundance Decreases in AD
*Bacteroidetes*	Firmicutes
*Faecalibacterium prausnitzii*	Clostridium sensu stricto 1
*Desulfovibrio*	
*Eubacterium rectale*	
*Porphyromonas gingivalis*	
*Lactobacillus rhamnosus*	

**Table 2 nutrients-16-02366-t002:** Selected drugs and their impact on the microbiome [7,50].

Drug	Effects on the Microbiome in Relation to the Pathogenesis of Alzheimer’s Disease
Combination of ampicillin, vancomycin, neomycin, metronidazole, amphotericin-B	Decrease in *Firmicutes* and *Bacteroidetes* (and changes in bacterial metabolites).
Vancomycin	Significantly altered gut microbiota composition (in neonatal vancomycin)
Streptozotocin and ampicillin	Disruption in the equilibrium of gut flora
Penicillin V	Altered gut microbiota composition
Antidepressant selective-serotonin reuptake inhibitors (SSRIs), sertraline, paroxetine and fluoxetine	Antimicrobial activity against Gram-positive bacteria (such as *Staphylococcus* and *Enterococcus*)
SSRIs (fluoxetine)	Induced depletion of cecal levels of Prevotella and Succinivibrio/reduction in Lactobacillus johnsonii and Bacteroidales
Tricyclic antidepressants (TCAs)	Prevention of the growth of gut pathogens, such as *E. coli*, *Yersinia enterocolitica*.

## Data Availability

Data sharing is not applicable as no datasets were generated or analysed during the current study.

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
