# Peer review of "Correlation between Alzheimer’s Disease and Gastrointestinal Tract Disorders"

_nutrients, 2024, doi:10.3390/nu16142366_

Round 1

Reviewer 1 Report

Comments and Suggestions for Authors

The review entitled” Correlation between Alzheimer’s disease and gastrointestinal 2 tract disorders” has been written well, however the reviewer thinks that few points should be added to further make it better.

Ø  Please add information about the duration onset of Alzheimer after H. Pylori infection.

Ø  Also add information about type of population and region, race, age specific to the Correlation between Alzheimer’s disease and gastrointestinal 2 tract disorders.

Ø  Is there any GI drug induced Alzheimer’s report? Add note in the reviews.

Ø  Add flowchart for literature review and studies selected. It will increase the understanding of selection of study and predict validity of correlation between Alzheimer and GI disorder.

Ø  Add exact number of case where Alzheimer and GI Disorder is common with references.

Ø  What is the treatment deficiency in treating H. Pylori which leads to Alzheimer.

Ø  Add a pictorial for brain synapses and show transaction between different neurotransmitters, indicating their levels decreasing or increasing lead to disease progression.

Ø  Specify studies to be conducted for future understand the concept, so that reader who are interested take your insights to plan study.

Author Response

Dear Reviewers, 

Thank you very much for your reviews. 

Authors have tried to address all of the issues that were suggested. 

These are the results listed in the order of your comments: 

Rev.1 

Ø  Please add information about the duration onset of Alzheimer after H. Pylori infection.

We added information about how many years after Helicobacter infection the highest risk of AD onset occurs. 

Ø  Also add information about type of population and region, race, age specific to the Correlation between Alzheimer’s disease and gastrointestinal 2 tract disorders.

We added information about age and sex specific to the correlation between AD and H.pylori infection in the introduction part.

Ø  Is there any GI (gastrointestinal) drug induced Alzheimer’s report? Add note in the reviews.

Scopolamine (drug used among others, in gastroenterology) has been used to induce Alzheimer’s disease in animal model studies. We found no reports of scopolamine induced AD in humans. We added information about scopolamine to the introduction part.

Ø  Add flowchart for literature review and studies selected. It will increase the understanding of selection of study and predict validity of correlation between Alzheimer and GI disorder.

We added a flowchart for literature review and studies selected.

Ø  Add exact number of case where Alzheimer and GI Disorder is common with references.

We added information about the number of cases of AD after Helicobacter pylori infection in the introduction section.

Ø  What is the treatment deficiency in treating H. Pylori which leads to Alzheimer.

We added information about Helicobacter pylori treatment deficiency and disadvantages.  

Ø  Add a pictorial for brain synapses and show transactions between different neurotransmitters, indicating their levels decreasing or increasing lead to disease progression.

New figure in Synaptic Dysfunction section added.

Ø  Specify studies to be conducted for future understand the concept, so that reader who are interested take your insights to plan study.

The need for further and new studies added in the “Conclusions” part.

In this version of the manuscript changes were marked in yellow. 

Yours faithfully, 

Authors of this manuscript 

Reviewer 2 Report

Comments and Suggestions for Authors

Ny suggestions: 

1. I would summarize the microbiomes in a table, which may impact the AD onset.

2. I think, a few figures may improve the manuscript further. For example in the case of Helicobacter. or Role of microbiome in Alzheimer's disease.

3. Besides Helicobacter pylori, can other bacteria influence intestinal imbalance and AD. 

4. Do AD  patients with mutations with APP, PSENs, or APOE E4 alleles have different gut microbiota patterns, compared to those without any genetic factors?

5. I would add a table on the drugs, which may impact the microbiota balance and could be a possible target to prevent neurodegeneration.

6. Instead of the Materials and Methods, I would add the manuscript's purpose to the end of the introduction. 

7. I would also change the chapter "Results" title to something else.

8. Does microbiome imbalance change gene expression, involved in neurodegeneration? The authors may add a few examples. 

Author Response

Dear Reviewers, 

Thank you very much for your reviews. 

Authors have tried to address all of the issues that were suggested. 

These are the results listed in the order of your comments: 

Rev. 2

  1. I would summarize the microbiomes in a table, which may impact the AD onset.

We added a summary of bacteria, which abundance increases in AD and has been linked to an increased risk of developing Alzheimer's disease and those which abundance decreases in AD. 

  1. I think, a few figures may improve the manuscript further. For example in the case of Helicobacter. or Role of microbiome in Alzheimer's disease.
  • New figure on role of microbiome in Alzheimer's disease added. 
  • New figure in Helicobacter pylori section added

  1. Besides Helicobacter pylori, can other bacteria influence intestinal imbalance and AD.

 New part about other bacteria added. 

  1. Do AD  patients with mutations with APP, PSENs, or APOE E4 alleles have different gut microbiota patterns, compared to those without any genetic factors?

We added information about APOE4 and PSENs mutations impact on gut microbiota. 

  1. I would add a table on the drugs, which may impact the microbiota balance and could be a possible target to prevent neurodegeneration.

A table on the drugs, which may impact the microbiota balance added. 

  1. Instead of the Materials and Methods, I would add the manuscript's purpose to the end of the introduction. 

We deleted the title “Materials and Methods” and transferred this part to the end of introduction. 

  1. I would also change the chapter "Results" title to something else.

We changed the term “Results” to “Selected gastrointestinal tract disorders”. 

  1. Does microbiome imbalance change gene expression, involved in neurodegeneration? The authors may add a few examples. 

We added a new section called “Microbiome imbalance and gene expression”. 

In this version of the manuscript changes were marked in yellow. 

Yours faithfully, 

Authors of this manuscript 

Round 2

Reviewer 2 Report

Comments and Suggestions for Authors

The authors fulfilled my suggestions. Thank you.